# The Impact of Action Planning after Causation-and-Effectuation-Based Entrepreneurship Education

**DOI:** 10.3390/bs13070569

**Published:** 2023-07-10

**Authors:** Khin Sandar Thein, Yoshi Takahashi, Aye Thanda Soe

**Affiliations:** 1Graduate School of Humanities and Social Sciences, Hiroshima University, Hiroshima 739-8529, Japan; d202094@hiroshima-u.ac.jp; 2Department of Commerce, Yangon University of Economics, Yangon 11041, Myanmar; ayethandasoe@gmail.com

**Keywords:** action planning, entrepreneurial self-efficacy, entrepreneurial intention, opportunity recognition, entrepreneurship education

## Abstract

The entrepreneurship literature shows inconsistent results in outcome effectiveness, such as entrepreneurial self-efficacy (ESE), entrepreneurial intention (EI), and entrepreneurial behavior. This could be due to the sole focus on the motivational aspects of behavioral change. Action planning, a volitional intervention used to modify health behavior, could resolve the inconsistent results mentioned above. Therefore, this study aims to evaluate the direct impacts of action planning intervention (API) following entrepreneurship education (EE) on ESE, EI, and opportunity recognition and to examine the behavioral change process from motivational and volitional perspectives. In this randomized controlled trial (RCT), we considered action planning to enhance entrepreneurial behavior after EE. The sample included 83 participants from a university in Myanmar. We randomly assigned the students to the API and control groups. We collected data on ESE and EI before and after training. We used objective measures for opportunity recognition through an opportunity evaluation framework. Serial mediation analysis revealed that the volitional impact on opportunity recognition was positively significant. From a motivational standpoint, ESE improved significantly, but we found no significant impact on EI; ESE and EI were serial mediators, with no specific mediation solely by ESE or EI. The findings contribute to the EE literature by presenting a brief and cost-effective API for EE.

## 1. Introduction

Entrepreneurs create new businesses in the face of risks and uncertainties to achieve profits and growth by identifying opportunities and procuring the necessary resources to capitalize on them [1]. Thus, education or training programs that promote entrepreneurship should not only try to impart the required knowledge and skills to students but also foster their entrepreneurial behavior. However, such programs lack attention to aspects such as improving cognitive learning and content retention, which are required for learners to build self-confidence and motivation. It can be challenging to identify the pedagogical factors (i.e., how and what to teach students) in EE programs that enable students to obtain the required knowledge and skills as well as to reach behavioral change goals.

Kirby [2] stated that the content and pedagogy of EE should focus on venture creation, creativity, and change, as well as helping students modify their behavior in order to become more entrepreneurial and to develop attributes such as role orientation for effectiveness, the ability to think both analytically and intuitively, and motivation to drive their actions [3]. Educators have begun using a pedagogy in which students are required to act and transform their ideas into market offerings through experiential learning [4].

However, Gunzel-Jensen and Robinson [5] pointed out that regardless of the ways in which EE provides experiential learning projects that create opportunities for practice, novice student entrepreneurs do not view them as real-life undertakings. Thus, Yamakawa et al. [6] suggested that EE pedagogy should include the practices of play, empathy, creation, and reflection so that students can experience the nature of running a business and put their thoughts into practice. Students must reflect on their experiences or knowledge and consider change as a natural state. Nabi et al. [7] also mentioned the need for a self-regulatory function that addresses uncertainty and setbacks.

Nabi et al. [7] made the connection between pedagogy and outcomes in EE. In particular, they pointed out that even in the context of pedagogy, which tries to alter entrepreneurial behavior, students do not seem to continue engaging in that behavior, despite their entrepreneurial intention (EI). Nabi et al. examined the gap in the literature regarding how intention has been transformed into behavior. Bayron [8] proposed the use of social cognitive theory (SCT) in EE and concluded that an intervention can enhance students’ cognitive factors and entrepreneurial behavior; however, there is still a need to apply SCT in the curriculum and in contexts of student interaction. SCT seems to explain students’ EE learning throughout the education process. However, SCT does not clearly distinguish which component affects behavior, and it has a minimal focus on motivation. Normally, in the training field, health action process approach (HAPA) models are utilized for health behavioral change interventions to reduce the intention-behavior gap by adding self-regulatory functioning. HAPA models explain the motivational and volitional phases of the behavioral change process; they are mostly used in the volitional phase [9]. While intention is formed in the motivational stage, volitional intervention provides a series of self-regulatory strategies to initiate and maintain a behavior [10]. Among the different HAPA models, action planning is composed of “what,” “when,” “where,” and “how” one will respond to situational cues. The addition of “how” is different from other models, such as those focused on implementation intention; this component is believed to include the motivational aspect [11]. As such, it would be interesting to study whether self-efficacy and intention can also be formed as motivational processes of behavioral change after an action planning intervention (API) that is independent of the volitional path.

In addition to EE programs, action planning, as part of post-training interventions, is expected to enable a more holistic implementation of EE, encouraging students to think both analytically and experientially while being trained to face risks and uncertainties in distinct situations. Hence, we aimed This study aims to evaluate the effectiveness of this additional API following EE on ESE, EI, and students’ opportunity recognition in Myanmar. This country’s economy has suffered not only from the impact of COVID-19 but also from other political and social conflicts, resulting in a drastic decline in GDP of 17.9% in 2021 [12] and the contraction of employment by 8.9% in 2022 [13]. In such adverse conditions, some people are forced to become self-employed after losing their jobs; this encourages entrepreneurship, according to emerging theories about underdogs [14].

Some researchers believe that the political, economic, and institutional climate has had a significant impact on entrepreneurial activity [15]. While EE research has explored the motivational and behavioral outcomes of EE based on theory and empirical tests have been conducted in ordinary contexts, little is known about countries facing adversity [16]. Karamti and Abd-Mouleh [17] proposed that individuals with similar qualifications from different contexts tend to react differently to the same opportunity. Thus, this study adds to the existing EE literature by examining the impact of EE on the motivational and volitional aspects of entrepreneurial behavioral change in adverse situations.

This study contributes to the literature by using SCT as a lens to evaluate training effectiveness and extends the literature from the perspectives of the motivational and volitional aspects by including action planning as a post-training intervention to improve the behavioral change process. We employed a randomized controlled trial (RCT) design in which students received action planning as a treatment intervention. This study also adds to the entrepreneurship literature by addressing the recommendation of prior scholarship that interventions to alter behavior should draw on theories of behavior and behavioral change. In sum, this study links pedagogy and students’ learning outcomes by designing a training program based on SCT theory, adding API from HAPA models as a post-training intervention, and presenting a sound mechanism to enhance individual behavioral change. We used RCTs to assess the effectiveness of the post-training intervention.

## 2. Literature Review and Development of the Hypotheses

### 2.1. Theoretical Foundations

An EE program is not just a training program but a complete process to help an individual become an entrepreneur. Alabduljader et al. [18] identified three goals of EE: (1) to increase students’ intention to start a new venture; (2) to foster students’ knowledge and skills to create a new business and manage it; and (3) to cultivate entrepreneurial characteristics and capabilities. EE aims to promote entrepreneurial behavior among students and graduates through the appropriate use of curricula and pedagogy [19,20]. An EE program should not only equip students with the necessary knowledge and skills but also help them become more entrepreneurial. Consequently, the outcomes of an EE program should be measured by the attainment of entrepreneurial behavior to ensure its effectiveness. 

Many entrepreneurship researchers have measured the effectiveness of EE using different outcome variables such as ESE, EI, and attitude. However, the literature has yielded inconsistent outcomes. In one study, ESE disappeared after the completion of training as a neutralizing effect [21,22], whereas other studies found a positive impact [23,24,25]. Further, in terms of intention, EE showed only a weak positive impact and/or intention—even declining over time [26]. Martin et al. [27] found a slightly positive effect, with other studies reporting a decreased effect [28], a non-existent effect [29], and even a negative impact [30]. However, other meta-analyses have found positive results for EI [31,32,33,34]. Although EI is considered to be the outcome of predictors such as ESE and attitude and also an impact of EE to increase the number of people starting a business, this incongruency in the literature highlights the need for more detailed analysis [35]. Therefore, we considered the formation of EI through ESE and how it transforms into behavior by applying API and SCT. The use of a stronger, theory-driven framework will hopefully resolve the contradictory findings regarding the impact of EE on outcomes, as suggested by Nabi et al. [7].

#### 2.1.1. The Role of SCT in EE

From a theoretical perspective, theories from social psychology have been the most influential in EE. Among them, the theory of planned behavior (TPB) by Ajzen [36] and the self-efficacy theory of Bandura [37], based on SCT, have been predominantly used in the field of entrepreneurship [38]. TPB has been widely used in the literature to predict entrepreneurial behavior, with EI being the best predictor [36].

While TPB is good at explaining the process of EI, the intention-to-action gap is a great weakness of the theory. Some authors believe there is a lack of detailed consideration of how entrepreneurs learn and a lack of substantial agreement on EE settings [39]. Particularly in the EE field, Bayron [8] pointed out the need to link learning and entrepreneurship and to teach entrepreneurship more effectively. He suggested the use of SCT in EE, concluding that the intervention can enhance students’ cognitive factors such as self-efficacy, which in turn influences entrepreneurial behavior; however, there is a need to apply SCT in curriculum development and student interaction efforts. Following his suggestion, SCT could be utilized not only to evaluate appropriate outcomes but also to design pedagogy so that they can be linked together to evaluate EE more effectively.

The essence of SCT is reciprocal interaction between individuals, the environment, and their behaviors. It is grounded in Bandura’s social learning theory, which focuses on social learning through the reinforcement and observation of past experiences from the environment, thereby building self-efficacy in personal internal processes to change behavior. SCT focuses on self-efficacy and self-judgment of one’s ability to perform a task in a specific domain [40]. The tripartite relationship between the environmental, personal, and behavioral components is enhanced through the development of self-efficacy.

Zhao et al. [41] illustrated the mechanisms of self-efficacy development through various pedagogical practices. They examined class activities and exercises as enactive mastery experiences, the inclusion of entrepreneurs as role models during lectures, case studies as vicarious experiences, social persuasion as a form of educators’ influence throughout a course, and the provision of lifestyles and work styles as examples during a course as students’ judgments of entrepreneurship. SCT seems to explain students’ learning effectiveness resulting from EE throughout the education process. However, one limitation of social cognitive models, such as TPB or SCT, is the unclear relationship between intention and behavior. Adding APIs can separate the motivational and volitional paths among the components of SCT and narrow the gap between intention and behavior.

#### 2.1.2. Action Planning in EE

Similar to many training programs, EE programs often successfully help individuals cultivate their cognitive abilities; however, they often fail to consider improving behavioral aspects. Training programs are usually targeted at achieving greater success in learners’ ability to transfer learning outcomes to the outside world by improving cognitive learning and content retention so that learners build self-confidence and motivation. Thus, entrepreneurship programs have failed to consider this aspect, although many have tried to facilitate entrepreneurial cognition. They provide the necessary knowledge and skills, yet they are still insufficient to equip learners with the motivation to transfer the acquired knowledge and skills to actual venture creation.

Social cognitive models have been used to explain people’s learning processes and the impact of EE. Among these, TPB and SCT are highly esteemed. TPB predicts the intention to perform a specific behavior. Ajzen suggested that intention is a strong predictor of behavior. Critics have pointed out the imperfect relationship between intention and actual behavior [42], suggesting intention is the motivation to perform a behavior. However, it remains unclear how behavior can be changed through pedagogical interventions. As such, we utilized action planning to extend SCT, making a clear distinction between the motivational path from ESE and EI to behavior and the volitional path leading directly to behavior. Behavioral change is enhanced by both motivation and volition [43,44,45].

Among the different intervention models, action planning is composed of motivational and volitional components; it has been used as an additional element of traditional social cognitive models [45]. With respect to training, Martin [11] proposed the use of follow-up activities to enhance learning transfer, which might not only be suitable for the workplace but could also help to change actual behaviors and build self-confidence after training. Of these activities, action plans are particularly aimed at specifying how trainees will implement learned skills on the job. Rather than the simple cue-to-action response in the usual health behavior change models (such as implementation intention as part of a volitional intervention), action planning includes cues such as the “when,” “where,” and “how” factors that respondents need to consider. These factors require effort as a self-regulatory process and the involvement of deliberate decision-making [45], reflecting the volitional and motivational aspects, respectively. It is worth differentiating between the motivational and volitional components of the behavioral change process in the field of training. While providing cognitive learning by helping learners become more knowledgeable about concepts and relationships within a given context, motivational and volitional components are also involved in action planning, as learners must describe what kinds of actions to take in what situations and how [46,47].

Requiring participants to create their own action plans encourages them to put the skills they have learned into practice while reducing their resistance to change [11]. Action plans serve as guidelines for trainees regarding what must be completed, with whom, and when, as well as how to implement learned concepts. As such, cognitive reasoning starts by recalling the models in the mind’s eye. Individual causal reasoning reacts to simulated contexts in which one is provided with hypothetical situations and performs if-then thought experiments [48]. In an educational setting, if-then thought experiments can be found in post-training interventions to help people retain their long-term memory and enhance desired behavioral outcomes [49,50].

Conventionally, EE programs—particularly those based on social cognition models such as TPB and SCT—have been targeted at motivational aspects (such as EI) to form entrepreneurial behavior. The provision of self-regulatory abilities, which are lacking in SCT [45], is missing in EE; this can be added through a volitional planning intervention. Similar to many other training programs, EE programs are offered and evaluated through psychosocial determinants (such as ESE and EI) for behavioral change. Action planning provides two paths for the behavioral change process: motivational and volitional. Along the motivational path, ESE and EI are targeted as the main outcomes, and volitional intervention is effective in enhancing the pick-up and retention of behavior [51]. To the best of our knowledge, this study is the first to include both motivational and volitional interventions, wherein students are required to keep in mind details regarding how they will perform when faced with certain situations throughout their post-intervention period until they form the desired behaviors.

### 2.2. Developing the Hypotheses

#### 2.2.1. Action Planning and ESE

Garcia and King [52] defined self-efficacy as the “self-appraisal of one’s ability to accomplish a task and one’s confidence in possessing the skills needed to perform that task.” Self-efficacy comes from SCT, which encompasses a tripartite relationship between the environment, personal feelings, and behavior [53]. It describes a person’s mental internal processes and interactions with others as a kind of reinforcement and observation. Four processes can influence self-efficacy: enactive mastery, role modeling and vicarious experiences, social persuasion, and judgments. Influencing the level of effort and persistence in performing a specific task can help people determine their unique choice of activities and behaviors [41]. Self-efficacy illustrates an individual’s auto-system for controlling his/her thinking, feelings, motivations, and actions [8].

In EE, ESE refers to the self-efficacy required to perform entrepreneurial tasks, which can be enhanced through the learning opportunities that individuals encounter during training [54,55,56]. By serving as guidelines for trainees on what must be completed, with whom, and when, as well as how to implement the concepts learned, action plans help them gain mental practice to rehearse a task in the absence of physical movement [57]. Mental practice is a type of mastery of experience, which is the strongest source for improving self-efficacy [58]. Morin [59] explained that mental practice enhances self-efficacy through three aspects of SCT: enactive mastery, vicarious experiences, and self-guided verbal persuasion. People who set action plans obtain mastery of their experience, which increases their self-efficacy [60]. Therefore, the use of action planning as a post-training intervention can enhance students’ ESE.

**Hypothesis** **1** **(H1):**
*An API following the main EE will have a positive impact on ESE.*


#### 2.2.2. Action Planning and EI

Using TPB as a reference theory, Krueger and Carsrud [61] initiated an EE research trend to determine its effectiveness in influencing the EI of students toward a greater likelihood of entrepreneurial behavior [62]. Although EI was found to be one of the outcome variables of EE, it ended up in the motivational stage and was unsure of moving on to the volitional phase. Normally, health behavioral change models are provided in the volitional phase to strengthen the intention-behavior relationship. However, the HAPA models’ volitional intervention was tested to examine the mediation process of psychosocial variables, such as self-efficacy and intention, after the formation of prior intentions [51]. If conventional EE could provide motivational aspects of EI, additional volitional intervention is again expected to increase intention by facilitating a clearer mental process with specific actions or strategies to be performed upon receiving situational cues, reactivating another motivational aspect, although its primary target is promoting behavior. Liang et al. [51] found a positively significant impact of the HAPA intervention on students’ intentions related to engaging in physical activity and consuming fruits and vegetables while controlling for the effect of self-efficacy as a mediator in the model. Therefore, we posited that action planning provided after effectuation-and-causation-based EE would enhance students’ EI. 

**Hypothesis** **2** **(H2):**
*An API after the main EE will have a direct and positive impact on EI.*


#### 2.2.3. Action Planning and Opportunity Recognition

For an EE program aimed at undergraduate students, it is difficult to measure students’ immediate entrepreneurial activity. Edelman et al. [63] defined the most important nascent entrepreneurial practices as “defining market opportunities, customers, and competitors” and “investing (one’s) own money.” For students in an ongoing undergraduate course, the first option of opportunity recognition can be the most suitable proxy variable for entrepreneurial behavior. Entrepreneurial skills such as observing and experimenting, questioning, and idea networking are behaviors for opportunity recognition [64], which can be considered part of entrepreneurial behaviors. Wasdani and Manimala [65] regarded the ability to recognize opportunities as a prerequisite for entrepreneurship and business ownership. According to Kuckertz et al. [66], entrepreneurial activities for opportunity recognition include being alert, actively searching for and gathering information, communicating information, addressing customer needs, and evaluating the viability of the abovementioned entrepreneurial activities. Hence, entrepreneurial activities for opportunity recognition can be regarded as proxies for entrepreneurial behavior.

Cognitive models predict behaviors through motivational aspects such as attitude, social norms, and perceived behavioral control, as in Ajzen’s TPB [51] and self-efficacy in Bandura’s SCT [53]. However, highly driven people might not necessarily act upon their intentions [49]. Kautonen et al. [67] suggested that entrepreneurship research must extend to behaviors rather than remain limited to intention. In other words, EE should be designed for behavioral enhancement, even though it cannot be measured immediately after a training program. 

Schwarzer [9] predicted that action plans would automatically generate behaviors through a cue-to-action relationship. Further, planning operates on volitional components and can be independent of intention [50]. Action planning clarifies the relationship between cues and response (behavior) [45] by encouraging students to consider “how” they respond to the “what,” “when,” and “where” aspects of situations. Therefore, embedding this post-training intervention, especially action planning in EE, would help increase entrepreneurial behavior in terms of opportunity recognition among students, even without the mediation of ESE and/or EI.

**Hypothesis** **3** **(H3):**
*An API after the main EE will have a direct, positive impact on opportunity recognition.*


#### 2.2.4. The Mediation Effect of ESE on the Relationship between Action Planning and Opportunity Recognition

Research has identified self-alertness, prior knowledge, self-efficacy, and social networks as antecedents of entrepreneurial opportunity recognition. Entrepreneurs with high self-efficacy are more innovative, and their economic performance is enhanced [68]. Park [69] suggested that self-efficacy is an important predictor of opportunity recognition. Ozgen [70] also pointed out the relationship between self-efficacy and entrepreneurial opportunity recognition. Chen et al. [54] argued that self-efficacy can predict a greater probability of exploiting opportunities because such activities require confidence to successfully implement a venture opportunity. Wang et al. [71] showed that high self-efficacy produced greater entrepreneurial opportunity recognition. Gibb [39] examined how entrepreneurial and creative self-efficacy can affect opportunity recognition and related behaviors. When developed through educational programs and training institutions, high self-efficacy can improve entrepreneurs’ innovative ideas [68], and they are better able to recognize opportunities. Particularly in the HAPA intervention for physical activity, self-efficacy was found to play a mediating role in controlling intentions [51]. 

Regarding H1, ESE is also expected to increase through action planning. Therefore, ESE is assumed to mediate the relationship between action planning and opportunity recognition.

**Hypothesis** **4** **(H4):**
*ESE serves as a mediator in the relationship between API and opportunity recognition among students.*


#### 2.2.5. The Mediation Effect of EI on the Relationship between Action Planning and Opportunity Recognition

Ajzen [36] predicted that intention is linked to action. This suggests that individuals with higher EI are more likely to engage in entrepreneurial behavior [62], serving as a motivational path in SCT. Although the effect size is different, EI can explain variations in action [67]. Kuckertz et al. [66] correlated opportunity recognition and exploitation with nascent entrepreneurial behavior. Action planning is also expected to have a motivational influence on EI, according to H2. Hence, EI is expected to serve as a motivational mediator between action planning and opportunity recognition.

**Hypothesis** **5** **(H5):**
*EI serves as a mediator in the relationship between API and opportunity recognition among students.*


#### 2.2.6. The Serial Mediation of ESE and EI on the Relationship between Action Planning and Opportunity Recognition

Action planning is composed of “what,” “when,” “where,” and “how” components; it is expected to pass through the self-regulatory process [72] and to require the deliberate processes of decision-making and self-evaluation. This intervention is expected to increase both motivational and volitional change [9]. In the field of entrepreneurship, ESE has been adopted in diverse entrepreneurial intention models based on SCT [73]. Intention has mostly been used as a mediator in behavioral change models, treated as a turning point between the initial goal-setting phase and the pursuit phase [9]. Self-efficacy also resembles perceived behavioral control and is used interchangeably or as an extension to predict behavioral intention [74]. Self-efficacy is required during the behavioral change phase to master different tasks. In a prior study, self-management cues and follow-up assessments performed after three weeks showed that self-efficacy and intention served as sequential mediators between intervention and dental-flossing behavior [9]. In another study, compared to the control group, participants in the intervention groups gained higher self-efficacy and increased their intentions for fruit and vegetable consumption and lifestyle changes [51]. Both self-efficacy and intention serve as mediators to alter behavior when planning interventions. By utilizing an API after the main EE training, entrepreneurship self-efficacy is expected to develop further, thereby increasing entrepreneurship intention to enhance opportunity recognition by students, showing the whole process of the motivational path from action planning to opportunity recognition.

**Hypothesis** **6** **(H6):**
*ESE and EI serve as partial serial mediators in the relationship between API and opportunity recognition among students.*


## 3. Methods

We conducted an RCT of an API after the main EE program to ensure a causal relationship in the mechanism of entrepreneurship learning. An advantage of the current study is that we identified a link between pedagogy, the research methodology, and the outcome variables.

### 3.1. Respondents

We announced that all 252 second- and third-year undergraduate students from the Department of Commerce at a university in Myanmar focused on business and economics could voluntarily participate in the Virtual Venture Incubating Program, an online entrepreneurship boot camp. We informed the students that they would be provided with compensation for Internet charges and a certificate upon completing the program. A total of 102 applicants, representing 40.48% of the total population, joined the training on the first day. On the last day, 93 students remained for the post-training intervention; we split them into two groups: 47 in the API group and 46 in the control group.

### 3.2. Intervention

The participants engaged in an intensive, 7-day online entrepreneurship workshop in which they learned both effectuation and causation techniques of entrepreneurship as part of the main EE. Using the causation approach, students are trained to conduct systematic analyses, develop strategies, and prepare a business plan. Effectuation principles require students or entrepreneurs to create business opportunities based on the means available to them, to determine the affordable loss for investment, to form partnerships with potential stakeholders, to act on contingencies, and to take control of all possible resources and actions. While causation logic is suitable for a predictable environment, effectuation principles require students to think like expert entrepreneurs. The use of the integrated approach of the two principles mentioned above would better equip students with the required knowledge and skills to think and act like entrepreneurs in different environmental contexts that are predictable and/or unpredictable. They can also develop ambidexterity for entrepreneurship. Hence, we provided integrated causation-and-effectuation training for the main EE.

The training took 10 days, with two hours per section per day for the main EE intervention. Upon completing the main EE, we randomly assigned the students to two groups: the API group or the control group. To reduce bias, we did not explicitly inform the students about the intervention; we segregated them into two breakout sessions, and we requested that the control group respond to the questionnaire with the help of a research assistant while the treatment group engaged in the post-training intervention. The treatment group received the API as the post-training intervention, which lasted for approximately 45 min, wherein students were asked to reorganize and recall what they had learned throughout the workshop as well as how to utilize the acquired knowledge and skills in stimulating situations with shared cues.

Moreover, students in the treatment group were asked to share in which of the following situations in their hypothetical company they believed they would be able to use their entrepreneurial skills:Research and development (R&D);New partnership contracts/forming alliances;Financing the capital required for product development;Changing the selling strategy;Supply chain logistics;Creating a new product.

They were then required to write at least five statements based on their situation: which skills they would use, when, where (i.e., specific situations), and how they would apply the knowledge and skills gained.

### 3.3. Data Collection

We collected data twice on demographic information, self-efficacy, entrepreneurial intention, and opportunity recognition (once before the training and immediately afterward) through a Google Forms survey composed of three sections (demographic information, ESE, and EI). For the objective measures of opportunity recognition, explain the opportunity evaluation framework to the students; they had to submit written assignments, which had to be evaluated by the researcher/trainer after five days of having completed the training. As this was an experimental training, we invited the students to voluntarily participate, and we obtained their consent through Google Forms that they themselves filled out. For ethical consideration, we also invited students from the control group to voluntarily participate in the API after they submitted their assignments. Of the 93 students, 47 joined the treatment group breakout session, while 45 of the 46 assigned to the control group participated in another breakout session. We did not explicitly inform the students about the post-training intervention, but we assigned them to two breakout sessions; the control group was led by a research assistant who was also a junior teacher from the university. We requested that students from the control group respond to the post-training survey, whereas those from the treatment group received the intervention.

### 3.4. Measures

For ESE, we used 18 subjective items from Kolvereid and Isaksen’s [75] study as they are based on the most influential measures of ESE that have been developed thus far. The dimensions included (1) opportunity recognition; (2) investor relationships; (3) risk taking; and (4) economic management. We did not consider marketing, human resources, or group interpersonal skills in this measure. Respondents had to indicate their degree of confidence in performing the various tasks successfully along an 11-point scale ranging from “no confidence at all” to “complete confidence” on each of the items. Sample items included “I see new market opportunities for new products/services” and “I could discover new ways to improve existing products/services”.

For EI, we used an 8-item subjective measure from Hoffman’s [76] study, with a 5-point Likert scale ranging from “strongly disagree” to “strongly agree.” We selected the items from various EI measures suitable for undergraduate students, as items reflecting the behavior of interest are defined in terms of the target, action, context, and time (TACT) elements described in a prior study [77].

For stronger evidence of the outcome evaluation, we used objective measures of opportunity recognition as the students submitted their assignments individually. We explained the opportunity evaluation framework of Winsor and Hanlon [78] to them; the students were required to identify opportunities and evaluate them based on features such as market needs, demand, market size, growth potential, duration of opportunity, competition, competitive advantage, industry structure and stage, resource requirements, human capital, and other entrepreneurial aspects.

### 3.5. Analysis

We performed a *t*-test and mediation analysis using Process Macros in SPSS. To gauge the outcome effect of the intervention between the treatment and control groups, as well as the ESE, EI, and other characteristics before intervention, we performed an independent samples *t*-test in SPSS 27 as a preliminary analysis to establish the total effects of the intervention. We employed serial mediation analysis via Hayes’ Process Macros in SPSS to test the hypotheses, both for the direct and three mediation paths.

## 4. Results

### 4.1. Descriptive Statistics of the Demographic and Pre-/Post-Intervention Data

This section describes the descriptive statistics of the respondents’ demographic traits in both the treatment and control groups, as well as the correlation matrix, along with the reliability value of each measure. Of the 83 final valid respondents, 42 were part of the treatment group, receiving the API after entrepreneurship training, while 41 were in the control group, receiving no additional intervention but only entrepreneurship training. Before testing the main analysis, we performed a balance check between the experimental and control groups using the likelihood chi-square test. We found no significant differences in terms of sex (χ^2^ = 0.002, *p* = 0.96) or family background (χ^2^ = 0.004, *p* = 0.95). An independent sample *t*-test of the age difference between the two groups was also not significant (t = 0.53, *p* = 0.59). Therefore, we judged the randomization to be successful.

The students’ average age was 19.20 years (ranging from 18–22 years of age). There were 12 male and 71 female participants in the overall sample. Regarding whether running a business was part of their family background, 36 students answered “no” and 47 students answered “yes.” Table 1 presents their demographic traits.

As one of our main goals was to determine the effectiveness of API in EE, we performed an independent samples *t*-test in SPSS to determine whether there were any differences in the demographic traits and pre-/post- measures of the outcome variables (such as ESE, EI, and opportunity recognition) between the treatment and control groups. Table 2 shows the *t*-test/chi-square statistics of the demographic and outcome variables before and after the intervention in the two groups.

Table 2 displays the results of the *t*-test. The mean values of pre-ESE taken before the training between the treatment group (*n* = 42, mean = 6.27, SD = 1.57) and the control group (*n* = 41, mean = 5.90, SD = 1.50) were not significantly different (t = 1.09, sig = 0.28). The pre-EI values between the treatment group (*n* = 42, mean = 3.95, SD = 0.63) and control group (*n* = 41, mean = 4.08, SD = 0.51) were also not significantly different (t = −1.08, sig = 0.28). However, after the training, the mean difference in the post-ESE results between the treatment group (*n* = 42, mean = 7.57, SD = 1.10) and control group (*n* = 41, mean = 6.28, SD = 1.17) was significantly different (t = 5.17, sig = 0.00, mean difference = 1.29), with higher ESE noted among students in the treatment group. Further, the mean opportunity recognition score of students from the treatment group (*n* = 42, mean = 61.21, SD = 11.64) was significantly higher than that of students from the control group (*n* = 41, mean = 54.02, SD = 10.72) and significantly different (t = 2.93, sig = 0.00, mean difference = 7.19). Unexpectedly, the post-EI results did not show any significant difference in the mean values of the treatment group (*n* = 42, mean = 4.17, SD = 0.50) or the control group (*n* = 41, mean = 4.22, SD = 0.41) (t = −0.50, sig = 0.62, mean difference = −0.14). 

### 4.2. Potential Confounders, Reliability, and Validity

As this model was based on SCT, we considered personal characteristics—such as age, gender, family business background, and the students’ own prior ESE and EI—to be potential confounders; we therefore tested these variables to see if there were correlations among them and with the main construct variables (such as post-ESE, post-EI, and opportunity recognition).

Table 3 shows the correlation matrix of the possible confounders and main construct variables. Personal demographic variables such as age, sex, and family background had no significant correlations with ESE, EI, or opportunity recognition. We did not include these demographic variables as control variables. However, individual personal attributes—such as pre-ESE (which had a positive correlation with post-ESE), post-EI, and pre-EI (which had a positive correlation with post-EI)—had very high potential to influence post-training data as potential confounders. Hence, we controlled pre-ESE and pre-EI in the mediation model.

When testing the reliability of the constructs (i.e., ESE and EI), the SPSS results showed that the Cronbach’s alpha value for ESE was 0.955 and for EI it was 0.834. Since both of the variables were above the standard criteria of 0.7, ESE and EI had good reliability. The reliability scores are displayed in brackets for each variable in Table 3. To test the validity, we performed factor analysis in SPSS for the constructs ESE and EI. To examine convergent validity, we calculated the value for the average variance extracted (AVE), and we found that ESE had an AVE score of 0.59 and that EI had an AVE score of 0.61. Both variables had AVE values above the minimum criteria of 0.5, such that the variables had convergent validity. To test discriminant validity, the AVE value has to be greater than the square of the correlation, which was 0.2025 (which is 0.45^2^). The two variables thus satisfied the condition of having discriminant validity as well.

### 4.3. Testing the Hypotheses

To test all motivational and volitional paths of action planning, including direct and indirect effects and sequential mediation analysis, we used Process Macros by Hayes [79] in SPSS 27. To test the serial mediation effect as shown in the model (Figure 1), the direct paths from X to Y (c’) needed to be significant. Next, to test the total effect, we examined the total indirect effect, which constitutes a long-way-specific indirect path from action planning to opportunity recognition through ESE and EI (a_1_db_2_) and two shortcut-specific indirect paths from X on Y through M_1,_ which is ESE (a_1_b_1_), and through M_2,_ which is EI (a_2_b_2_). The serial mediation effect required the long-way-specific indirect path to be significant. We used Process Macros from Model 6 to test the model’s direct, indirect, and total effects. As possible confounders, we added pre-ESE and pre-EI as control variables since they could influence the training outcomes.

Please note the following symbols in the model:

X: action planning

M_1_: ESE

M_2_: EI

Y = opportunity recognition

a_1_db_2_ = long-way-specific indirect path from X to Y through M_1_ and M_2_

a_1_b_1_ = shortcut-specific indirect path from X to Y through M_1_

a_2_b_2_ = shortcut-specific indirect path from X to Y through M_2_

c’ = direct effect of X on Y

Total indirect effect = a_1_db_2_ + a_1_b_1_ + a_2_b_2_

Table 4 shows the regression outcomes of the mediation model. The first model, in testing the effect of action planning on ESE, revealed a positively significant result (b = 1.24, *p* = 0.000), with R^2^ = 0.48. Providing action planning after the main EE can build students’ ESE. Thus, H1 was supported. In the second model, the direct effect of action planning on EI was negatively significant (b = −0.19, *p* = 0.03), with an R^2^ value of 0.60. Action planning may even reduce students’ intentions to start a business. Thus, H2 was not supported. In this model, ESE also had a significant positive effect on EI (b = 0.17, *p* = 0.00). Students with higher ESE displayed a greater intention to start a business. However, in the full model, ESE did not have a significant effect on opportunity recognition (b = 0.03, *p* = 0.94). Even though the students had high ESE, it did not necessarily lead to opportunity-recognition behavior. Again, EI had a positively significant impact on opportunity recognition (b = 8.11, *p* = 0.05). Students with higher EI demonstrated better opportunity recognition behaviors. The direct effect of action planning on opportunity recognition was also positively significant (b = 7.89, *p* = 0.01); action planning had a considerably large influence on enabling students’ opportunity recognition behavior. H3 was therefore supported due to the significant impact of the direct volitional path. The full model had an explanatory power of 19%, R^2^ = 0.19, and the total effect of action planning on opportunity recognition was high (b = 8.20, *p* = 0.01). When we added mediators to the full model, the coefficient fell to 7.89, but it was still significant. According to Table 5, the pathway from action planning to opportunity recognition was explained only by the long-way indirect path through ESE and EI (b = 1.72, LLCI = 0.15, ULCI = 3.74) but not by ESE alone (b = 0.04, LLCI = −3.43, ULCI = 3.50) or EI alone (b = −1.44, LLCI = −3.55, ULCI = 0.01). Hence, H4 and H5 were not supported. Nonetheless, the long-way indirect effect was significant, supporting H6.

## 5. Discussion and Implications

### 5.1. Discussion

Our main purpose was to determine the effectiveness of including API in causation-and-effectuation-based EE as well as to understand the behavioral change process of students as the underlying mechanism. Based on SCT, we developed a framework for EE and tested the impact of additional action planning in EE on the motivational and volitional aspects of behavior. 

In our framework, the direct impact of additional action planning after EE can be categorized into the following: the impact on motivational factors (such as ESE and EI) and the volitional aspect of change in the nascent entrepreneurial behavior of opportunity recognition after controlling for the motivational factor-based paths. H1 tested the direct motivational impact of action planning in EE; the results from the serial mediation model of Process Macros showed that the API had a positively significant impact on ESE as a motivational influence. By enhancing mental practice through action planning as an enactive mastery process, our study confirms the findings of Domke et al. [60] and Warner et al. [58], with the motivational impact of API on ESE in EE literature particularly. However, for H2, another motivational direct impact of action planning on EI was negatively significant, unlike the original expectation. Even though students’ confidence improved, it did not influence the students’ EI directly after controlling for ESE. Even though the hypothesis was not supported, the results seem realistic. One possible justification for this outcome is that due to the students’ ability to build a clear mental picture, they became more realistic and realized the challenge of running their own business, thereby decreasing their false hope of intending to start one. As studies on the effect of API on ESE or EI are new, there is no previous literature available for direct comparison. However, considering action planning as part of EE, our results may be related to the findings of Bae et al. [31], who conducted a meta-analysis, revealing that the effect of EE on EI was insignificant, including the mediation effect of ESE, and those of Krueger and Carsrud [61] and Fayolle and Moriano [80]. In the current study, the direct effect on EI, after controlling for mediation, showed a significantly negative outcome due to action planning. H3 examined the volitional direct effect of API on opportunity recognition. As expected, we found a positive direct effect by controlling the motivational paths.

In our study, we divided the behavioral change process, as an effect of action planning on behavioral change, into two aspects: the motivational and volitional paths. Assuming the behavioral change process to be part of the enactive mastery process, based on H3, we expected action planning to improve opportunity recognition behavior directly under volitional paths. On the other hand, from a motivational standpoint, we expected ESE and EI to mediate the relationship between action planning and opportunity recognition independently according to H4 and H5 or serially via ESE and EI according to H6. The estimation outcomes of the serial mediation model indicated that neither ESE nor EI served as mediators independently, as expected in H4 and H5, respectively, for the relationship between action planning and opportunity recognition in their separate estimations. The unexpected result for H4 (a_1_ × b_1_) is due to the insignificant direct effect of ESE on opportunity recognition (b_1_). One possible reason is the potential influence of the situational factors facing the students. During a crisis, experienced entrepreneurs with high ESE can recognize opportunities better than those with low ESE [17]. However, the undergraduate students, who had no prior experience with entrepreneurship, found it difficult to locate good opportunities in such vulnerable situations, even if they had high ESE. Moreover, another unexpected outcome for H5 (a_2_ × b_2_) is due to both the negative effect of action planning on EI (a_2_) and the insignificant effect of EI on opportunity recognition (b_2_). The justification for the former is elaborated for H2. In terms of the latter, we can again assume the presence of situational factors, such as a crisis. In this context, even though the students would have liked to run their own businesses in the future, this motivation was insufficient to work on their own opportunity recognition. However, as in H6, they sequentially formed a motivational process because the serial mediation effect was positively significant. Although the third path (b_2_) was individually insignificant at the 5% level (*p* = 0.054), the insignificance was quite marginal. Hence, according to the bootstrapping estimation outcome, we found the serial motivational effect of action planning through ESE and EI to lead to better opportunity recognition. Although action planning is normally used to address the intention-behavior gap, it enhances the mastery process, thereby increasing ESE and EI and activating motivational paths.

Beyond the interpretation of the results for each hypothesis, we wished to compare the motivational and volitional path-based effects for further discussion. As mentioned above, with regard to the paths from the API toward opportunity recognition, the direct effect (c’) and the serial mediation effect (a_1_ × d × b_2_) were significantly positive, while mediation via ESE (a_1_ × b_1_) and mediation via EI (a_2_ × b_2_) were insignificant. Consequently, the volitional path-based effect (c’ = 7.89) was more than four times as large as the sum of the mediation path-based effect (1.72 = insignificant (a_1_ × b_1_) + insignificant (a_2_ × b_2_) + 1.72 (a_1_ × d × b_2_)). Although we did not make any speculations on the relative strengths of the two paths when developing the hypotheses, judging from the main objective of action planning, the result is likely to be acceptable.

### 5.2. Theoretical Contribution

Using SCT, we conducted an API to enhance students’ learning and entrepreneurial performance in terms of ESE, EI, and opportunity recognition. SCT describes the interactions between three elements: the environment, the person, and behavior. The use of integrated effectuation and causation alone is expected to improve students’ cognitive ability; however, to strengthen entrepreneurial behavior, we added an action planning component as a post-training intervention in this study. Although APIs have been used for health behavioral change models, we explored changes in ESE, EI, and opportunity recognition as proxies for nascent entrepreneurial behavior to understand the behavioral change process. As expected, ESE and behavior improved. When designing a curriculum and pedagogy, educators should consider how action planning can be added as a stimulus to create a clearer mental picture. This study also reminds educators and policymakers of the importance of being entrepreneurial rather than solely focusing on the number of startups created or one’s intention to start a business. In order to spur the economy or uplift it from a crisis, organizations need to be more entrepreneurial. Managers and employees should also be ready to support the opportunity recognition process of the organization as intrapreneurs.

### 5.3. Practical Implications

Apart from API, when considering the main content of EE, most undergraduate students will not have prior business experience, and it will be rather difficult for them to act as expert entrepreneurs, as in the case of effectuation. Therefore, most nascent entrepreneurs tend to think more causally, whereas effectuation is linked to realistic solutions and better behavioral outcomes. To resolve this challenge, although adding an API takes less time than the main EE, it can help students build a clearer mental picture. For educators, regardless of how they build their course content, the API component can influence students’ behavioral change in a positive manner.

Further, when planning an EE course, the impact evaluation needs to consider whether students have truly achieved the objectives, and it needs to be designed properly to measure the objectives after the program. Once all necessary components (such as course content, objectives, pedagogy, and additional support) are well linked, the success of an EE program will be more pronounced, and socioeconomic improvement will be supported in the future.

## 6. Conclusions

To overcome the weaknesses of most EE impact evaluation studies, based on Nabi et al.’s [7] recommendations, we attempted to address prior concerns related to methodological and statistical issues, to include pedagogical interventions, and to present a stronger theory-driven framework. We used both effectuation and causation as the baseline approaches to EE, as they would be considered the most effective in fulfilling the objectives of EE. Further, to ensure that EE could really change students’ performance or help them become entrepreneurs, we used API to provide students with the opportunity to reflect on what they had learned throughout the course in a specific situation. This study contributes to the literature by analyzing the impact of action planning through the motivational and volitional paths.

To confirm causality in terms of impact evaluation as well as how students experience the learning process, we utilized RCT to present the API after the main EE program. Hence, this study contributes to the EE impact evaluation literature through its methodological rigor. With an RCT study design and pedagogy carefully designed and aligned with the outcome variables, our results can address contradictory findings in the EE impact literature, such as those by Bae et al. [31], and we can respond to Martin et al.’s [27] call for a more thorough research design.

To understand the learning process and to design a pedagogical approach, we used SCT as a baseline theory, and we used an API as the post-training intervention from health behavioral change models to enhance learning in SCT. Meanwhile, we selected related outcomes such as ESE, EI, and opportunity recognition as outcome variables; we thus attempted to match all important research elements such as objectives, content, pedagogy, outcomes, and methodology, thereby contributing to the literature on EE.

## 7. Limitations and Suggestions for Future Research

We utilized an RCT and found a causal effect of EE on the outcome variables with a careful pedagogical design grounded in a strong theoretical framework. However, this study has certain limitations. First, we administered the survey to undergraduate students, and we could not measure the long-term impact of the intervention on them. Instead, we could only use opportunity recognition as a nascent entrepreneurial behavior of Level 3, in addition to the Level 2 outcomes of Nabi et al. [7]. Longitudinal studies are highly recommended to confirm long-term causality. Second, the participants’ average age was fairly young (mean = 19 years), even for undergraduate students, and the total sex ratio was largely influenced by female participants due to the country’s education system and the university’s historical enrollment status. Schmidt et al. [81] found that male and female participants showed differences in ESE and opportunity recognition. However, we did not consider sex differences. Third, we did not consider family background in business in the model, as family background in entrepreneurship positively impacts entrepreneurial intention [82].

A major factor to consider is Myanmar’s political and economic situation facing the students. While COVID-19 negatively affected the country’s economy, additional conflicts in the political and social environment harmed the economy, employment status, and social welfare. The decrease in EI in the results also reflects this situation. We highly recommend implementing future studies in other countries to confirm the learning process and impact of the intervention. Nevertheless, this study encourages the implementation of a post-training intervention after EE and is expected to accelerate the momentum of the EE literature.

## Figures and Tables

**Figure 1 behavsci-13-00569-f001:**
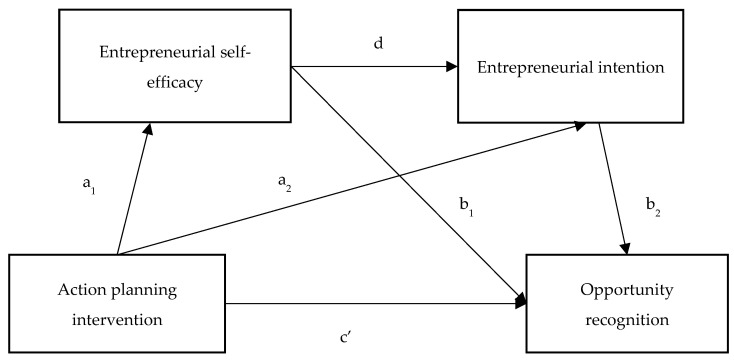
Analytical model of the study.

**Table 1 behavsci-13-00569-t001:** The respondents’ demographic traits.

	Frequency	Percentage
Sex		
Male	12	14.5%
Female	71	85.5%
Age		
18	37	43.6%
19	8	9.6%
20	28	33.7%
21	9	10.8%
22	1	1.2%
Family background: Having a business		
Yes	36	43.4%
No	47	56.6%

**Table 2 behavsci-13-00569-t002:** Independent sample *t*-test/chi-square test of the demographic data and main construct variables.

	Total		Action Planning		Control		Independent Sample *t*-test/χ^2^				
	(*n* = 83)		(*n* = 42)		(*n* = 41)						
	Mean	SD	Mean	SD	Mean	SD	t/χ^2^	df	*p*	95% CI	
1. Age	19.20	1.20	19.19	1.08	19.05	1.32	0.53	81	0.59	−0.39	0.67
2. Sex	1.86	0.35	1.86	0.35	1.85	0.36	0.00	1	0.96		
3. Family background	0.58	0.50	0.55	0.50	0.55	0.50	−0.62	1	0.51		
4. Pre-ESE	6.09	1.54	6.27	1.57	0.61	0.49	−1.08	81	0.28	−0.30	1.04
5. Pre-EI	4.02	0.57	3.95	0.63	4.08	0.51	1.09	81	0.28	−0.39	0.11
6. Post-ESE	6.94	1.30	7.57	1.10	6.28	5.17	5.17	81	0.00	0.79	1.79
7. Post-EI	4.20	0.46	4.17	0.50	4.22	0.41	−0.50	81	0.62	−0.25	0.15
8. Opportunity recognition	57.66	11.70	61.21	11.64	54.02	10.72	2.93	81	0.00	2.30	12.08

**Table 3 behavsci-13-00569-t003:** Correlation matrix.

	1	2	3	4	5	6	7	8	9
1	1								
2	0.09	1							
3	0.10	−0.08	1						
4	0.05	0.01	−0.09	1					
5	−0.04	−0.01	0.10	−0.12	1 (0.959)				
6	0.08	0.03	0.09	0.50 **	0.51 **	1 (0.835)			
7	0.08	−0.03	0.12	−1.12	0.42 **	0.30 **	1 (0.952)		
8	0.14	−0.03	0.05	−0.06	0.24 *	0.45 **	0.69 **	1 (8.874)	
9	0.20	0.19	0.04	0.21 **	−0.01	0.25 *	0.12	0.27 *	1

Note: 1 = age, 2 = sex, 3 = family, 4 = action plan, 5 = pre-ESE, 6 = post-ESE, 7 = pre-EI, 8 = post-EI, 9 = opportunity recognition. *p*-value is significant at * < 0.05 and ** < 0.01. Cronbach’s alpha value in (xxx).

**Table 4 behavsci-13-00569-t004:** Regression coefficients, standard errors, and model summary information for the serial multiple mediator model (n = 83).

	ESE				EI				Opportunity Recognition			
		Coeff.	S.E	*p*		Coeff.	S.E	*p*		Coeff.	S.E	*p*
Action planning	a_1_	1.24	0.22	0.00	a_2_	−0.18	0.08	0.03	c’	7.89	3.00	0.01
ESE					d	0.17	0.04	0.00	b_1_	0.03	1.47	0.94
EI									b_2_	8.11	4.18	0.05
Constant		2.58	0.78	0.00		1.56	0.26	0.00		25.42	11.44	0.03
	R^2^ = 0.48				R^2^ = 0.60				R^2^ = 0.19			
	F(3,79) = 24.34, *p* = 0.00				F(4,78) = 29.02, *p* = 0.00				F(5,77) = 3.62, *p* = 0.01			

Note: Coefficient, unstandardized regression coefficient; SE, standard error; *p*, *p*-value; n, number of participants.

**Table 5 behavsci-13-00569-t005:** The effect of action planning, ESE, and EI on opportunity recognition.

			B	SE	95% Confidence Interval	
					LLCI	ULCI
Total effect		c	8.21	2.50	3.24	13.17
Direct effect		c’	7.89	3.00	1.91	13.87
Indirect effect	via ESE only	a_1_ × b_1_	0.04	1.75	−3.43	3.50
	via EI only	a_2_ × b_2_	−1.44	0.95	−3.55	0.01
	via ESI and EI	a_1_ × d × b_2_	1.72	0.91	0.15	3.74

## Data Availability

The data presented in this study will be openly available at Mendeley Data; https://doi.org/10.17632/5xjf2br8t5.1.

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
