# Peer review of "The Impact of Action Planning after Causation-and-Effectuation-Based Entrepreneurship Education"

_behavsci, 2023, doi:10.3390/bs13070569_

Round 1
Reviewer 1 Report
The reviewed article focuses on the effectiveness of action-planning intervention after entrepreneurship training and its impact on entrepreneurial self-efficacy, entrepreneurial intention, and opportunity recognition. The study utilized a t-test and mediation analysis using the process macro in SPSS. I find the idea of the paper interesting and to the point. There are however, some points which require some improvements.
The article lacks a clear justification for choosing a t-test and mediation analysis as the primary statistical methods. The rationale behind these choices and their appropriateness for the research questions should have been explained in more detail.
The article also does not provide sufficient information about the data collection process. Details such as the sampling method, response rate, and measures taken to ensure data validity and reliability are missing. This information is crucial for assessing the quality and generalizability of the findings.
The discussion section of the article is relatively short and lacks in-depth analysis of the findings. The authors could have provided a more comprehensive interpretation of the results, discussed the practical implications, and highlighted any limitations or potential confounding factors that might have influenced the outcomes.
Reviewer 2 Report
Thanks for the opportunity to review the article, I would like to bring some notes as below:
1. I find the abstract a bit confusing not clearly able to deliver a clear understanding of the objectives of the study and how it is presented.
2. Please check the definition of EE in the first two lines of the article, it seems like something is missing i.e., “direct and indirect education of students and would-be entrepreneurs in the art and science of new venture and value creation”.
3. Authors have mentioned the importance of two approaches for teaching entrepreneurship and EE, I find it interesting but still I find it challenging to understand them more clearly. Please elaborate more on them to allow readers understand the two approaches and their differences, you should understand that not all readers are experts in your area hence things should be clear to them.
4. I believe the introduction is too lengthy, some part of it can come in the literature review section.
5. There should be a section for the theoretical background and there you can put your theory.
6. Please have a separate section for literature review and hypotheses development, don’t combine everything together.
7. Please enrich your study with this recent article (Factors Influencing Entrepreneurial Intention of University Students in Yemen: The Mediating Role of Entrepreneurial Self-efficacy, 2023).
8. Please provide if you have any demographic information for the respondents.
9. Please separate the study implications from the conclusion section
10. Please focus more on the context of the study in the introduction section.
11. Please compare your findings with some more previous studies.
12. The literature review for ESE needs more elaboration, please refer to this source and adopt it or follow the structure of writing (Green Innovation, Self-Efficacy, Entrepreneurial Orientation and Economic Performance: Interactions among Saudi Small Enterprises, 2023).
13. Please draw your own tables instead of using the SPSS directly.
14. The article needs some proofreading as there are some areas where English issues exist.
All the best
NA
Round 2
Reviewer 1 Report
I am pleased to acknowledge that the revisions have significantly enhanced the quality and overall value of the article. The adjustments made have effectively addressed the areas of concern and have contributed to a more refined and compelling piece of work. In my opinion, the article is now in a state where it is suitable for publication.
Reviewer 2 Report
satisfied